# Estimation of Hg(II) in Soil Samples by Bioluminescent Bacterial Bioreporter *E. coli* ARL1, and the Effect of Humic Acids and Metal Ions on the Biosensor Performance

**DOI:** 10.3390/s20113138

**Published:** 2020-06-02

**Authors:** Irena Brányiková, Simona Lucáková, Gabriela Kuncová, Josef Trögl, Václav Synek, Jan Rohovec, Tomáš Navrátil

**Affiliations:** 1Institute of Chemical Process Fundamentals of the Czech Academy of Sciences, Rozvojová 135, CZ-16502 Prague 6, Czech Republic; branyikova@icpf.cas.cz (I.B.); simona.lucakova@vscht.cz (S.L.); kuncova@icpf.cas.cz (G.K.); 2Department of Biotechnology, University of Chemistry & Technology, Prague, Technická 5, CZ-16628 Prague 6, Czech Republic; 3Faculty of Environment, Jan Evangelista Purkyně University in Ústi nad Labem, Králova Výšina 3132/7, CZ-40096 Ústí nad Labem, Czech Republic; vaclav.synek@ujep.cz; 4Institute of Geology of the Czech Academy of Sciences, Rozvojová 269, CZ-16500 Prague 6, Czech Republic; rohovec@gli.cas.cz (J.R.); navratilt@gli.cas.cz (T.N.)

**Keywords:** bioluminescent bioreporter, mercury detection, pollution bioavailability, *Escherichia coli* ARL1, whole-cell biosensor

## Abstract

Mercury is a ubiquitous environmental pollutant of dominantly anthropogenic origin. A critical concern for human health is the introduction of mercury to the food chain; therefore, monitoring of mercury levels in agricultural soil is essential. Unfortunately, the total mercury content is not sufficiently informative as mercury can be present in different forms with variable bioavailability. Since 1990, the use of bioreporters has been investigated for assessment of the bioavailability of pollutants; however, real contaminated soils have rarely been used in these studies. In this work, a bioassay with whole-cell bacterial bioreporter *Escherichia coli* ARL1 was used for estimation of bioavailable concentration of mercury in 11 soil samples. The bioreporter emits bioluminescence in the presence of Hg(II). Four different pretreatments of soil samples prior to the bioassay were tested. Among them, laccase mediated extraction was found to be the most suitable over water extraction, alkaline extraction, and direct use of water-soil suspensions. Nevertheless, effect of the matrix on bioreporter signal was found to be severe and not possible to be completely eliminated by the method of standard addition. In order to elucidate the matrix role, influences of humic acid and selected metal ions present in soil on the bioreporter signal were tested separately in laboratory solutions. Humic acids were found to have a positive effect on the bioreporter growth, but a negative effect on the measured bioluminescence, likely due to shading and Hg binding resulting in decreased bioavailability. Each of the tested metal ions solutions affected the bioluminescence signal differently; cobalt (II) positively, iron (III) negatively, and the effects of iron (II) and nickel (II) were dependent on their concentrations. In conclusion, the information on bioavailable mercury estimated by bioreporter *E. coli* ARL1 is valuable, but the results must be interpreted with caution. The route to functional bioavailability bioassay remains long.

## 1. Introduction

Mercury is a global pollutant affecting health of living organisms as well as entire ecosystems. Although acute mercury poisoning is nowadays rare thanks to legislation requirements and monitoring, the scale of chronic exposure to lower doses of Hg as a result of a global pollution or an occupational hazard is still extensive [1,2,3].

As mercury itself cannot be decomposed or degraded, it is constantly cycled and recycled through a biogeochemical mercury cycle. This includes not only the transfer of mercury between different environments, e.g., atmosphere, fresh and seawater, soils, sediments, biosphere, etc., but also chemical interchanges and changes in the strength of mercury’s bonds to the surrounding matrices. The most important primary inputs to the mercury cycle are burning of anthropogenic-fossil fuels and artisanal gold mining. In the case of mercury, “input” means release of mercury from non-available bonds and locations to more available forms [4].

In the atmosphere, mercury occurs as (i) Hg(0) vapors with relatively low reactivity, (ii) highly reactive mercury compounds, e.g., HgCl_2_, and (iii) particle bound mercury [4]. The gaseous mercury can be, among other depositions, assimilated by the tree leaves and conifer needles [5,6] and consequently deposited via leaf drop to the soil surface. Surface deposition is more frequent means of mercury input to plants than root uptake [7].

In soils, the mercury is generally not very mobile, except in microbial volatilization [7]. As mercury is readily adsorbed by organic material, the highest mercury concentration (up to 1 mg/kg) usually occurs in organic soil horizons with high humus content [8]. Due to the order of magnitude of lower soil organic material, mercury concentration in mineral soils typically reaches up to 100 µg·kg^−1^ [9]. With respect to organic carbon, mercury is released slowly from the soil organic material during decomposition (humification) of soil organic material. Also, the tillage of agricultural land accelerates oxidation of soil organic material, resulting in low mercury content of agricultural soils, e.g., in the Czech Republic up to 60 μg·kg^−1^ [10].

In general, biogeochemistry of mercury in soils has been dominated by inorganic and organic mercury (II) complexes. Mercury adsorption to mineral and organic surfaces is mainly determined by pH and dissolved ions e.g., increasing Cl^−^ concentration and/or decreasing pH may decline Hg adsorption [11]. Due to strong affinity for reduced sulfur groups, Hg^2+^ and MeHg^+^ ions are readily bound to soil organic material and metal sulfides. Thus, in terrestrial unpolluted soil, Hg distribution has been primarily determined by abundance of soil organic material and/or clay [11]. A much less abundant specie in suboxic unpolluted soils has been Hg^0^; however, its importance increases in reduced environments and polluted soils [12]. In soil polluted by Hg mining, its speciation was typically dominated by occurrence of cinnabar (HgS) [13].

In freshwater, the mercury content is usually very low (1−10 ng/L). Similar to the situation in soil and atmosphere, the mercury in water environment binds to molecules of dissolved organic carbon or to particles. Elevated filtered Hg concentrations (~20 ng·L^−1^) were reported from streams typical with high dissolved organic carbon concentration [14]. Nevertheless, these concentrations are minor with respect to Czech drinking water legal limits of 1 μg/L [15,16,17]. During the periods of high water flow (e.g., during floods, snow melting, etc.), when sediments and soil particles are suspended in water, the concentrations of mercury in water increase by several orders [18]. An important phenomenon in the mercury cycle is the methylation of mercury. In anoxic environments, anaerobic bacteria bioaccumulate mercury and catalyze its methylation, transforming mercury to the most hazardous mercury component methylmercury (MeHg). Through the food chain, methylmercury may be concentrated (up to six orders of magnitude), so the highest amounts can be found in predatory fish such as swordfish, tuna or shark [19].

The following methods for mercury analysis have achieved sub-ppb detection limits: CVAAS (Cold Vapor Atomic Absorption Spectroscopy) and CVAFS (Cold Vapor Atomic Fluorescence Spectroscopy). Other widely used analytical methods include ICP-MS (Inductively Coupled Plasma Mass Spectrometry), FIA-AFS (Flow Injection Atomic Fluorescence Spectrometry), CVAES (Cold Vapor Atomic Emission Spectroscopy), Raman probe, Raman scattering and Anodic stripping voltammetry [20,21]. Although these advanced methods of analytical chemistry are highly precise and extremely sensitive, they don’t provide reliable information on mercury bioavailability and associated toxicity. For example, mercury in cinnabar (HgS) is highly stable and non-available; opposite to this is HgCl_2_, which is highly reactive and toxic. Moreover, it depends on the nature of the matrix, how tightly is Hg(II) bound and potentially bioavailable. Speciation and fractionation of mercury in the sample provides valuable information, but it is still not sufficient for the decisive prediction of the total toxicity [22] since different organisms can have different mechanisms of mercury enter. This complete task may be solved using whole-cell biosensors, which by their nature detect solely the bioavailable compounds [23]. On the other hand, the specificity and accuracy of the bioreporters are inferior compared to chemical analysis, as will be discussed later in this paper.

The legislation of the Czech Republic in compliance with the European Union guidelines establishes three different limits for the presence of mercury in agricultural soil. The first is a so-called “preventive value” of 0.3 mg/kg; exceeding this shall result in preventive measures that stop the further increase of the mercury concentrations. The second is the “indication value, above which the safety of food or feed may be threatened”, which is 1.5 mg/kg; such soil cannot be used for food and feed production. The third limit is the “indication value, above which the human and animal health might be threatened”, which is 20 mg/kg; solely contact with such a soil is considered to be dangerous for health of people or animals [15,16]. Due to the nonexistence of a reliable and precise method of assessment of bioavailable mercury, the legal limits are based solely on assessment of the total mercury content, even though the toxicity of different mercury species varies greatly. Therefore, the development of an analytical tool capable of bioavailable mercury assessment is urgently required.

Bioluminescent bioreporters are regarded to be useful for fast and inexpensive detection of bioavailable contaminants [24,25,26,27]. In previous years, we worked on the mercury detection by whole-cell optical bioluminescent bioreporter *Escherichia coli* ARL1 in fresh and saline water. We successfully developed a method of mercury estimation in water samples. The limit of detection was 20 ng/L for tap water and 570 ng/L for diluted laboratory sea water (25 vol.% of laboratory sea water in distilled water) [28,29].

This paper logically continues with this topic and is focused on the assessment of bioavailable mercury in soils samples. As the soils are generally extremely complex matrices and they contain a large number of compounds, which affect the bioreporter response, this task is highly demanding and has not yet been sufficiently fulfilled. Therefore, in this follow up paper we summarize results of experiments aimed at the detection of bioavailable mercury in real contaminated soil samples and at comparison of instrumental analyses results with the bioreporter response. Based on previous successful mercury detection in solutions, we focused predominantly on methods of mercury transfer to water. We tested measurements (i) directly in soil suspensions, (ii) in water extracts, (iii) in alkaline extract (in order to dissolve the organic matter more efficiently), and (iv) laccase assisted extraction (to facilitate dissolution of organic matter and associated mercury release). For a better understanding of the matrix effect on the performance of the bioreporter we used model solutions with humic acids, chemically pure representative of the soil organic matter, and selected metal salts (Fe^2+^, Fe^3+^, Ni^2+^, Co^2+^) representing ions possibly leached from soil inorganic components.

## 2. Materials and Methods

### 2.1. Chemicals and Solutions

All used chemicals were commercial products in p.a. grade. Distilled water was used for the preparation of all the solutions. NaCl, HgCl_2_, FeSO_4_·7H_2_O, FeCl_3_·6H_2_O, Fe_2_(SO_4_)_3_, NiCl_2_·6H_2_O, CoCl_2_·6H_2_O, MnCl_2_·4H_2_O, ZnCl_2_, citric acid monohydrate, trisodium citrate dihydrate (Lach-Ner, Neratovice, Czech Republic); NaOH, HCl, KH_2_PO_4_, Na_2_HPO_4_ (Penta, Prague, Czech Republic); tryptone (Oxoid, Basingstoke, UK), yeast extract, kanamycin sulphate, D-glucose, Humic acid (Sigma-Aldrich, St. Louis, MO, USA, No. 53680), Laccase from *Trametes vesicolor* (Sigma-Aldrich, St. Louis, MO, USA, No 38429, EC 1.10.3.2) were used in experiments.

Phosphate buffer (PB) (pH 7.4) contained KH_2_PO_4_ (0.23 g/L) and Na_2_HPO_4_ (0.74 g/L). Phosphate buffer saline (PBS) (pH 7.4) was prepared by supplementation PB with NaCl (0.15 mol/L). Citrate buffer (pH 4.4) contained citric acid monohydrate (1.0 g/L) and trisodium citrate dihydrate (1.6 g/L).

D-glucose stock solution (2 mol/L) was sterilized by filtration through a syringe filter (pore size 0.22 µm; Millipore, Millipore, France). Kanamycin stock solution (10 g/L) was sterilized by filtration through a syringe filter (pore size 0.22 µm; Millipore, Molsheim, France). Luria–Bertani media (LB) contained tryptone (10 g/L), yeast extract (5 g/L) and NaCl (10 g/L) was sterilized in autoclave (20 min, 121 °C, 205 kPa), final pH was 7.2. The tryptone solution (20 g/L) was sterilized by autoclaving (20 min, 121 °C, 205 kPa).

Mercury standard solution was prepared by dissolving HgCl_2_ in distilled water. Concentration of stock solution was 1 g/L Hg(II), lower concentrations were prepared by its dilution with distilled water.

Standard reference material (powder) IAEA/V-10 HAY (International Atomic Energy Agency, Vienna, Austria) certified value of the total mercury concentration 0.013 (0.009–0.016) mg/kg.

Humic acid stock solution–0.01 g of humic acid was dissolved in 0.625 mL of 0.2 M NaOH and filled up to 9 mL. Then it was neutralized with HCl and filled up to 10 mL, so a colloidal brown-colored solution was formed.

### 2.2. Microorganism–Bioluminiscent Bioreporter

The *mer-luxCDABE*-based genetically engineered bacterial bioreporter *Escherichia coli* ARL1 (EC100 [30]) was kindly donated from the collection of microorganisms of CEB University of Tennessee, Knoxville, TN, USA. The bioreporter produces bioluminescence in the presence of divalent mercury ions (Hg^2+^).

### 2.3. Soils

Seven soil samples (A, B, C, D, E, F, G) used for bioluminescent mercury estimation (Table 1) were collected in different areas of the Czech Republic in order to cover wide range of soil properties (organic matter content, mercury content and mercury speciation).

#### 2.3.1. Samples A to C

Soil samples A–C originate from pristine forest catchment Lesní potok 40 km east from the capital Prague and represent genetic soil horizons (details on sampling and site can be found in [9]). Sample A represents the forest floor (O-horizon) and was sampled from the surface using 15 cm × 15 cm frame. Samples B and C represent genetic soil horizons Ah and C and were sampled from an excavated 80-cm deep soil pit. Four modifications of soil samples A and C (A1, A2, C1, C2) were prepared by doping with HgCl_2_ to achieve soil combining high total carbon and high Hg(II) content (A1, A2) and low carbon and high Hg(II) content. Sample D originates from a historical Fe-ore and cinnabar mining site Jedová hora 50 km south from the capital Prague (details on sampling and site can be found in [13]). It was sampled from a 85-cm deep soil pit representing Bw genetic soil horizon. In brief, soils were freeze-dried, sieved through a 2 mm nylon mesh and stored in polyethylene containers.

The total content of mercury in soils was assessed by cold-vapor atomic absorption spectrometry with preconcentration of mercury by amalgamation on gold (CV-AAS, Hg analyzer—AMA 254, Altec, Prague, Czech Republic). The method is not a conventional CV-AAS, but method adapted to soil samples [31]. The determinations were performed in triplicate; the relative standard error for each sample mean was 3%—a measure of the measurement precision). Total carbon was determined using a Carlo-Erba 1108 elemental analyzer—Carlo Erba Instruments, Milan, Italy (the relative standard error 3.5%). The individual values of the standard errors for each result are given in Table 1.

#### 2.3.2. Samples E to G

Set of urban soil samples E–G were collected in the city of Ústí nad Labem around the former amalgam electrolysis plant (in operation 1928−2015); details are described elsewhere [32]. The samples were collected from the upper 15 cm, around 1.0−1.5 kg from each locality. Samples were air-dried and sieved through a 2 mm sieve. Mercury content was determined in dry samples by cold-vapor atomic absorption spectrometry with preconcentration of mercury by amalgamation on gold (CV-AAS, Hg analyzer—AMA 254, Altec, Prague, Czech Republic). The determinations were performed in triplicate, the relative standard error for each sample mean was 3%. CRM IAEA/V-10 HAY-certified value of total Hg = 0.013 (0.009–0.016) mg/kg was applied in internal quality control; mean of 4 measured values was 0.011 mg/kg, standard deviation (intermediate precision) was 0.001 mg/kg. The organic matter was estimated as the loss at 375 °C after previous drying at 105 °C, the result was calculated as the mean of two repeated determinations, the relative standard error 3.5%.

### 2.4. E. coli ARL1 mer-lux Bioassay

The bioassay consists of four subsequent steps (Figure 1):Sample preparation (Section 2.4.1)Preparation of *E. coli* ARL1 culture in exponential growth phase (Section 2.4.3)Induction and recording of bioluminescence in *E. coli* ARL1 (Section 2.4.4)Evaluation of measured data (Section 2.4.5)

To eliminate the effect of pH on the bioluminescence, buffer was used where possible (for bacterial suspension and for laccase mediated extraction).

#### 2.4.1. Sample Preparation

The soils were homogenized in the agate mortar grinder (Alfred Fritsch, Germany, bowl diameter 10 cm, pestle diameter 3.5 cm, 40 rpm, 60 min, sample weight 20 g), the resulting particle size was estimated by sedimentation to be within the range of 10−100 μm. Each of the 11 ground soils (Table 1) was subjected to four following ways of sample preparation (Section below).

*Soil suspensions.* Soil samples were mixed with distilled water to form suspension and vortexed for 1 min. Based on preliminary tests, during which soil concentrations from 2 g/L to 20 g/L were tested, the concentration of 4 g/L was chosen to be the most suitable regarding signal linearity and intensity (data not shown). The suspensions were directly used for the induction of bioluminescence. Alternatively, the standards of mercury (II) chloride were added to suspensions prior to the induction of bioluminescence so that final concentrations of added Hg^2+^ were 1; 5; 25 and 50 µg·L^−1^.

*Water extractions.* A quantity of 0.1 g of each soil was placed in 50 mL Erlenmeyer flasks, 10 mL of distilled water was added, flasks were shaken on a rotary shaker at 150 rpm, 24 h, 20 °C, subsequently the suspensions were centrifuged at 2264× *g*, 10 min, supernatant was collected and used for the induction of bioluminescence. Alternatively, the standards of mercury (II) chloride were added to supernatants prior to the induction of bioluminescence so that final concentrations of added Hg^2+^ were 1; 5; 25 and 50 µg·L^−1^.

*Alkaline extractions.* A quantity of 0.1 g of each soil was placed in 50 mL Erlenmeyer flasks, 10 mL of 0.2 M NaOH was added, flasks were shaken on a rotary shaker at 150 rpm, 12 h, 20 °C, subsequently the suspensions were centrifuged at 2264× *g*, 10 min, supernatant was collected and another 10 mL of 0.2 M NaOH was added to the sediment and shaken for another 12 h. After centrifugation, supernatants were combined and used for the induction of bioluminescence. Alternatively, the standards of mercury (II) chloride were added to the combined supernatants prior to the induction of bioluminescence so that final concentrations of added Hg^2+^ were 1; 5; 25 and 50 µg·L^−1^.

*Laccase mediated extraction.* To decompose the humic substances in soils to release bound mercury and avoid the dark color, the enzyme laccase was used. Laccase from *Trametes versicolor* EC 1.10.3.2 was diluted in citrate buffer (pH 4.4) in concentration 1 g/L (>500 U/L). 50 mg of soil sample was placed in 25 mL Erlenmeyer flask, 10 mL of laccase buffer solution was added and the suspension was mixed with magnetic stirrer (10 mm, 600 rpm, 25 °C) for 24 h [33]. Suspension was centrifuged at 2264× *g*, 10 min, and the supernatant was used subsequently for the induction of bioluminescence. Alternatively, the standards of mercury (II) chloride were added to the supernatants prior to the induction of bioluminescence so that final concentrations of added Hg^2+^ were 1; 5; 25 and 50 µg/L.

#### 2.4.2. Matrix Effects

A series of experiments using only chemically defined substances (without soil) were done with the aim to evaluate an effect of further components of soil samples (humic acid and inorganic ions) on bioreporter performance.

*Effect of humic acid on the bioreporter performance*. To evaluate the effect of humic acid, the stock solution was diluted with the distilled water to get four humic acid concentrations: 1000; 100; 10 and 0 mg/L. Each solution was divided and standards of mercury (II) chloride were added to get eight different mercury concentrations: 0; 12.5; 25; 50 and 100 μg/L of Hg^2+^ (in total 32 combinations of humic acid and mercury concentrations). 

*Effect of inorganic ions on the bioreporter performance.* Solutions of metal ions were prepared in the concentrations based on the composition of the certified reference soil Orthic Luvisol S-MS 12-1-08 (pb-anal, Slovakia). The following metal salts and concentrations were chosen in order to cover the range from zero to the maximum concentration of metals in soil sample:Fe^2+^: FeSO_4_·7H_2_O, concentrations of 0; 75; 150; 300; 600 and 1200 mg/L of FeFe^3+^: both Fe_2_(SO_4_)_3_ and FeCl_3_·6H_2_O, concentrations of 0; 75; 150; 300; 600 and 1200 mg/L of FeNi^2+^: NiCl_2_·6H_2_O, concentrations of 0; 0.5; 1 and 2 mg/L of NiCo^2+^: CoCl_2_·6H_2_O, concentrations of 0; 0.5; 0.9; 1.8; 3.7; 7.4; 14.7 and 29.5 mg/L of Co

The solutions of the highest concentrations were used as stock solutions, which were subsequently diluted with distilled water; standards of HgCl_2_ in the concentration range of 0 to 20 μg/L of Hg(II) were added alternatively. The bioluminescence was induced according to Section 2.4.4.

#### 2.4.3. Preparation of *E. coli* ARL1 Culture in Exponential Growth Phase

Sterile 300 mL Erlenmeyer flasks closed with bacteriological plug containing 100 mL of LB media were inoculated with *E. coli* ARL1 from agar plate. Kanamycin stock solution was added in amount of 0.5 mL to get final concentration of 50 mg/L. The flask was placed on a rotary shaker and incubated overnight at 37 °C, 200 rpm. Overnight culture (3 mL) was inoculated into a fresh LB medium with kanamycin and incubated at the same conditions as the overnight culture to the approximate absorbance of A_600_ = 0.35 (~0.6 × 10^8^ CFU·mL^−1^) in order to get bacteria in exponential growth phase. OD was determined in 1 cm cuvette at 600 nm by UV-VIS spectrophotometer HP8452A (Hewlett-Packard, Palo Alto, CA, USA). The bacterial cells were centrifuged for 10 min at 2264× *g* (Hettich Universal 32R, Andreas Hettich, Tuttlingen, Germany), the pellet was resuspended in PBS with glucose (40 mmol·L^−1^) and tryptone (10 g/L) to a cell concentration of 1.2 × 10^8^ CFU·mL^−1^. This suspension was used for bioluminescence induction as soon as possible (within less than 10 min).

#### 2.4.4. Induction of Bioluminescence in *E. coli* ARL1

A volume of 125 µL of *E. coli* suspension, 125 µL of sample (Section 2.4.1) and alternatively in cases of “standard addition” 0 to 12.5 µL of mercury standard solution (Section 2.2) were filled in each well of a 96 well microtiter plate (Costar, white polystyrene plate, sterile, Corning Incorporated, Corning, NY, USA). The microplate was covered with gas permeable sealing membrane for microtiter plates (Diversified Biotech, Dedham, MA, USA) and incubated in SPECTROstar Omega microplate reader (BMG Labtech, Ortenberg, Germany) at 37 °C and double orbital shaking for 16 h. During the incubation, bioluminescence intensity as well as optical cell density for each well were recorded automatically every 12 min 37 s, for each well. The cell density was measured as the light absorbance at the wavelength of 600 nm. Bioluminescence intensity was measured in relative light units (RLU) in range 240−740 nm. The time series of the absorbance and bioluminescence were the outputs of the measurements.

#### 2.4.5. Data Evaluation and Statistics

Statistics were evaluated using Microsoft Excel Professional version 2019 and DataFit 8.1.69.

For each sample, the recorded time-course of the bioluminescence had always typical shape known for other bioreporters [34,35] (more detailed data are presented in Appendix A) with one maximum approximately 1.5 h after the beginning of experiment as can be seen in Figure 2. Both the maxima of bioluminescence and the integrated bioluminescence over the time period of 16 h were evaluated. Better linearity was found for the integrated values, so all the calculations presented in this paper are based on them. As soils are highly complex matrices, which itself influence the response of the biosensor, bioavailable mercury contents of the soils were estimated after different treatments (Section 2.4.1) by the method of standard addition. A plot was constructed for each sample as a dependence of integrated bioluminescence on the mercury concentration added. The trendline for this dependence was assessed by the linear regression. The original concentration of mercury in the sample was determined from the point at which the extrapolated trendline crosses the concentration axis at zero signal. The original mercury content of each sample was calculated from the trendline equation as follows:$$\mathrm{Trendline}\;\mathrm{equation}\;\mathrm{is}\;\int_{0}^{16h}L=m\cdot (c_{Hg}+c_{Hg,0})+b$$
$$\mathrm{for}\;c_{Hg}=0\;\;\;\;\mathrm{and}\;\;\;\int_{0}^{16h}L\equiv 0\;\Rightarrow \;c_{Hg,\;0}=-\frac{b}{m}$$
where *L* is bioluminescence (RLU—relative luminescent unit) $\int_{0}^{16h}L$ is integrated bioluminescence (RLU.h), *m* is trendline slope (-), *c_Hg_* is concentration of added mercury in the sample (μg/L), *b* is y-intercept of the trendline, *c_Hg_*_,0_ is the original concentration of mercury in the sample.

The runs were repeated three to five times. The standard error or standard deviation of the mean as an indicator of the measurement precision was computed from the sample standard deviation quantifying the variability of the mean values obtained in repeated runs and from the given replication number. In case of the analytical procedures with soil suspensions, laccase extractions and water extractions, the relative standard deviations (RSD) corresponding to the sample standard deviations are quite high (however corresponding to deviations of previous bioreporter bioassays [34,36]): for bioactive Hg concentration >0.04 mg/kg the average relative standard deviations were 91% (from 33% to 165%), 54% (from 36% to 91%) and 72% (from 38% to 123%), respectively; for low Hg concentration (<0.04 mg/kg) the standard deviations were <0.2 mg/kg. The results from the repeated runs were tested to detect outliers by Q- test; neither outlier was identified in the data at α = 0.01.

Correlations between all pairs of measured quantities were investigated using Spearman’s rank correlation coefficient. In the case of estimations of bioavailable mercury, the input data were the means of the results obtained from repeated runs.

The relationship between the bioavailable mercury contents and the total mercury contents were also investigated using weighted regression. 

## 3. Results

Bioluminescence responses to standard additions for selected samples and evaluation of linearity are presented on the Figure 2 and Figure 3a,b. The total mercury concentrations assessed by CV-AAS and the bioavailable mercury concentrations estimated by the biosensor are compared on the Figure 4. Table 2 presents correlations between Hg concentrations determined by means of instrumental analyses with concentrations estimated by bioluminescence response.

### 3.1. Bioluminescence of E. coli ARL1 in Soil Samples

All the soil samples were subjected to four different ways of pretreatment prior to the induction of bioluminescence: soil suspension preparation, water extraction, alkaline extraction and laccase mediated extraction. In most of the cases, the soil samples induced lower bioluminescence comparing to the solutions of HgCl_2_ in distilled water with the same mercury concentration. In Figure 4, all the results are summed up excluding the alkaline extraction, which was found to be overall unsuitable (Section 3.1.3).

#### 3.1.1. Bioluminescence Induction in Soil Suspensions

As the simplest way, bioluminescence was induced directly in the soil suspensions. In Figure 2, the time courses of the measured bioluminescence are plotted for the case of the suspensions for soil sample A and the four different standard additions. This illustrates the method of measurement; the respective plots for the other following cases are not presented, as they had very similar shape. The bioluminescence was integrated over the time period of the entire measurement for each sample and these values where used for the subsequent calculations. Dependence of the bioreporter signal on the mercury concentration given by standard additions for chosen soil samples is presented in Figure 3a,b. Figure 4a,b depicts contents of the bioavailable mercury estimated using the biosensor related to the total mercury content in soil samples assessed by CV-AAS with constant experimental conditions. For the majority of soil samples, the biosensor estimated values were significantly lower comparing to the total ones and they represent 1% to 73% of the total value. The only exception was the sample of the soil labeled C, where the biosensor estimated value was almost 20 times higher comparing to total mercury content. No correlation was found between the bioreporter response and TOC content or mercury content determined by instrumental analyses (Table 2).

#### 3.1.2. Bioluminescence Induction in Water Extracts of Soil Samples

Bioluminescence induction in water extracts was further attempt to estimate the bioavailable mercury content in soil samples. Unfortunately, the estimated values seem to be more or less independent of the total mercury content. In some cases the values were overestimated and in others underestimated in up to two orders of magnitude, see Figure 4c,d.

#### 3.1.3. Bioluminescence Induction of Alkaline Extracts of Soil Samples

As mercury is bound mostly on an organic soil matter (humic substances), which dissolve in alkaline environment, the soil samples were extracted by 0.2 M NaOH. Even though the extraction efficiency was found to be 70−80% for all of the samples excluding the soil sample labeled D (checked by instrumental mercury assessment in extracts, data not shown), this approach has not been found to be appropriate from following reasons. (i) The extracts were of high pH (above 12), so for the use of living whole-cell bacterial bioreporter, it was necessary to neutralize it. At neutral pH, dark black-brown precipitates were formed, so it was very difficult to take a representative sample. (ii) The extracts of the samples containing higher TOC (all samples except samples C, C1, C2) were of dark black-brown color, so no bioluminescence was recorded due to shading. (iii) It is very likely that high pH caused precipitation of Hg(II) in form of HgO. (iv) Thus, the alkaline extraction itself influenced the bioavailability of mercury significantly.

#### 3.1.4. Bioluminescence Induction of Laccase Extracts of Soil Samples

Laccase was used to decompose humic substances in soil samples partially in order to release bound mercury and to avoid dark color, which was a property of the alkaline extract (Figure 4e,f). This approach was the most successful; the estimated values represented 34 to 113% of the total mercury content of the samples A, B, E, F, G, C1, C2, A1, A2 respectively (Table 1). The only two samples with larger deviation were samples C, where the mercury content was overestimated 42 times and the sample D, where the estimated value represented only 3% of the total content.

### 3.2. Influence of Humic Acid on the Growth and Bioluminescence of E. coli ARL1

Humic substances are essential components of soil. In order to elucidate the effect of humic acid on the physiological state of the bacteria *E. coli* ARL the bioluminescence was induced in the colloidal solutions of different concentrations of humic acid (0; 10; 100 and 1000 mg/L) and HgCl_2_ (0; 12.5; 25; 50 and 100 μg/L of Hg(II)). The higher the humic acid concentration, the lower the measured bioluminescence for the same mercury concentration (Figure 5). On the other hand, the higher the humic acid concentration, the higher the growth rate of the bioreporter (Table 3). The biggest difference was between the cultures without humic acid (≤0.15 h^−1^) and the cultures with (≥0.19 h^−1^).

### 3.3. Influence of Metal Ions on the Bioluminescence of E. coli ARL1

The presence of different metal ions was found to substantially influence the response of the bioreporter (Figure 6a–d). The metal ions present in soil had very low or no ability to induce bioluminescence of the bioreporter itself without the presence of mercury. Nevertheless, when present together with mercury, they affected the bioluminescence greatly, some of them positively (Co^2+^), others negatively (Fe^3+^). Moreover, in cases of Fe^2+^ and Ni^2+^ the effect was dependent on the added metal concentration. For example, for constant concentration of mercury (II), concentration of nickel 0.5 mg/L decreased the bioluminescence of bioreporter, in opposite both lower and higher concentrations increased it. It has to be mentioned that at low concentration the Fe^2+^ ions are likely dominantly oxidated to Fe^3+^, therefore the Figure 6b,d are very similar at this area. These results show contradictory effect of metal ions on Hg^2+^-induced bioluminescent response of strain ARL1. Therefore, elimination of the effect of the sample matrix on the test results needs a more complex approach than frequently used comparing with constitutive-signal-emitting control cells.

## 4. Discussion

### 4.1. Relation of Bioreporter Response to Mercury Concentration

Overall the responses of the bioreporter did not correspond to Hg concentrations determined in soil by means of instrumental analytical methods. This is summarized in Table 1 (statistical evaluation of the most successful procedure, the laccase extraction) and Table 4 (statistical evaluation of weighted linear dependences between total Hg concentrations and Hg estimation by bioreporter). The weighed regression was used due to the data heteroscedascisity. Assuming linear dependence, we tested whether the slopes of Hg concentration determined instrumentally vs. those estimated by bioreporter are significant (i.e., question slope = 0 in Table 4), whether this slope corresponds to equality (i.e., slope = 1 in Table 4), and whether the intercept is significant (i.e., intercept = 0 in Table 4).

Considering responses to single concentrations (Table 3), part of the bioreporter responses were insignificant (i.e., they could not be distinguished from 0 based on *t*-test). Nevertheless, the entire concentration-response dependences were distinctly significant (i.e., the slopes were significantly >0, Table 4) but it was necessary to exclude two questionable results (see Table 4b).

If all samples were included in the regression than for laccase extraction and water extraction the slopes were insignificant, i.e., the bioluminescence response is not dependent on the total mercury concentration. Since in the case of samples C and D the bioluminescence responses were disproportionate to the total Hg concentrations, in contrast to the other samples (see Table 1), the results measured on them were excluded from the regression analysis (they form only a minor proportion of the total number and their different behavior can be explained—see below). Consequently, the slopes became distinctly significant for all three extraction methods (Table 4b), i.e., the relationship between the bioluminescence responses and the total Hg concentration was proved. Moreover, Table 4 shows that all slopes of all extraction methods were significantly lower than 1. The bioreporter thus generally underestimates the total concentrations.

Nevertheless, the bioreporters are believed to response to the bioavailable proportion of pollutant only. Bioavailability can be defined as a portion of pollutant, which is weakly bound, easily extractable or reactive; thus, as a consequence, it is able to enter and/or affect a living cell [37]. Assessment of the bioavailable portion of metal pollutant in soil samples using instrumental analyses is not an easy task and requires speciation and sequential extractions [23,38,39]. A living bacterial bioreporter might be simplification of the task, but it can provide valuable information, even though we do not measure directly a portion of mercury entering the bacterial cell, we can estimate it from the measured signal (bioluminescence).

Beside the response to a bioavailable proportion of pollutant only, the bioreporters have their selectivity of response [34,40]. As far as we know, no exhaustive study was carried out to determine the response of the ARL1 bioreporter strain to different mercury species. Since the ARL1 strain is based on the *mer* operone used naturally for detoxification of Hg^2+^ ions, the ARL1 should respond dominantly to Hg^2+^ species [30]. Speciation of the mercury forms was far beyond the scope of this study; our key aim was the comparison of different extraction procedures on real samples. Nevertheless for samples labeled as D we know that vast majority of present was in the form of very stable and insoluble cinnabar HgS, which is also a non-available form [13]. Elimination of these samples from the set resulted in improvement of the instrumental vs. bioreporter dependence. Overall this comparison supports the hypothesis that bioreporter ARL1 only responds to the bioavailable mercury proportion. Regarding the uncertainties of the linear dependences, the response shall only be considered semi-quantitative.

### 4.2. Mercury ARL1 Bioassay and Its Interferences

Bioreporters are living organisms which require an entire undisturbed metabolism to provide reliable results. The bioluminescence response can be especially negatively influenced by insufficient oxygen and various bactericide agents [41]. On the other hand, the bioluminescence increase (form of hormesis) can be observed according to some physiological changes, such as membrane disturbances or growth-phase [41,42,43]. These have to be considered when evaluating the bioreporter response. This very likely happened in the case of sample C, which possesses very low mercury concentration; however, the bioluminescent response was high. A clear effect of the matrix on the bioreporter response can be also observed from the comparison of deviations received upon standard addition of Hg(II) (which was done after the extraction process, Figure 3) in comparison to higher deviation observed from the response to repeated samples (including the extraction process).

In compliance with the literature [44], the bioluminescence induced in distilled water was significantly higher (and also less variable) than any of the soil samples (suspension or leachate). Subsequent experiments suggest that there are two main reasons for this phenomenon. (i) There is an effect of light absorption or scattering by soil particles and/or light absorption by dissolved humic substances as was discussed by Rasmussen [44], but (ii) there is also a significant effect of other metal ions presented in samples on the response of the biosensor, which was not considered by authors of the previous papers. We tested metal concentrations adjusted to those in the selected reference material (Orthic Luvisol S-MS 12-1-08) and observed that each had a large influence on the bioluminescence of the biosensor (positive, negative or concentration dependent). Regarding the fact that the concentrations of the metals are greatly variable in the different soils, it is hardly possible to predict the overall influence of a soil matrix on the biosensor response. Nevertheless, the soil extracts in general will most likely decrease the bioluminescence of the biosensor due to the relatively high concentration of iron (III) in most of the soils [45].

Humic acid had positive effect on the growth rate of *E. coli* ARL1 (Table 3). On the other hand, the measured bioluminescence was significantly lower in samples with humic acids. The higher the concentration of humic acid, the lower the bioluminescence for the same concentration of mercury. This effect could be caused by two mechanisms. Either the humic acids adsorbed the mercury ions and made them less bioavailable for the bioreporter, or/and the dark colored humic acid solution absorbed substantial part of the emitted light, so the measured bioluminescence was lower for the samples containing humic acid. Lower bioavailability of the Hg(II) ions would be connected not only with the lower bioluminescence induction, but possibly also with lower toxic effect of the mercury on the *E. coli* ARL1. The decrease of Hg^2+^-induced bioluminescent response of strain ARL1 in the presence of humic acids was ascribed to reduction of bioavailability of Hg^2+^ via complexation with HA also by Du et al. [46]. The comparison of the growth rates (Table 3) helped us to elucidate the humic acid action. As the growth rate differences among the cultures of different mercury concentrations without humic acid were almost undetectable, we can assume that the mercury is not toxic for the bioreporter in given concentrations. This assumption is supported by the *lux-mer* construction which uses the detoxification-elimination mechanism. Low toxicity of complexes with humic acids to *E. coli* was already shown by unchanged light emission of constitutive bioreporter *E. coli* 652T7 [46]. As a result, the positive effect of humic acid on the bacterial growth cannot be explained by protecting the cells from the toxicity of mercury by its binding. A positive effect of the humic substances on the bacterial growth was also observed by Tikhonov et al. [47] for most of the tested soil bacteria due to their capability to utilize parts of humic acid as a source of carbon.

The bioavailable mercury content for the soil sample C was estimated to be 20 to 42 times higher via bioreporter than the total Hg content determined by CV-AAS, which is obviously nonsense. Nevertheless, it should be noted that instrumentally the mercury concentration in this sample was near the limit of quantification and we can therefore consider very high uncertainty of the determined value. The effect of the soil matrix will also be considered. Modulation of bioreporter bioluminescence, both positively and negatively, without induction of reporter genes was described previously [41,42].

The best performance from all the tested soil sample preparations was obtained by the laccase extraction (Table 1). Even though laccases have wide industrial use [48], their application as leaching enhancers is unique. This approach developed by the authors facilitated the leaching procedure by partial decomposition of lignin under mild conditions, so that the extracts were suitable for subsequent bioluminescent assay. Unfortunately, even by this approach, it was not possible to reliably determine whether the given sample exceed or fulfill the legal criteria of preventive value (0.3 mg/kg) or indication values (1.5 mg/kg, 20 mg/kg [15,16]). Generally, the samples of low mercury content tend to be overestimated and the samples with high mercury content tend to be underestimated. Considering that natural samples of soil contain dozens of metal ions in unknown concentration and different amounts of organic matter, it is more or less impossible to describe or predict the mercury concentration of the soil samples solely from the bioluminescence, even with employment of standard addition method. On the other hand, the *mer-lux* bioassay could represent a valuable, simple, and relatively cheap screening method, when used for large number of samples taken in one area of similar soil type or it could serve as an additional information to the total mercury content of the soil.

### 4.3. Application of the Results and Future Perspectives

Bioreporters were first utilized for easy monitoring of the metabolism [49] or microbial toxicity and physiology [50,51]. Later attempts were carried out to use them in immobilized state for construction of optical biosensors [36,52,53,54], especially for water samples. Overall the limits of detection as well as selectivity could not be compared to instrumental analyses [28,34]. This was also confirmed by this study, where the significance of bioreporter responses were low and mercury estimations did not correspond to concentrations determined by means of instrumental methods. Applied to contaminated soils, the strength of the bioreporters should be the response to bioavailable proportion of the pollutant, which is a parameter important in applications (estimation of proportion of biodegradable pollutant or pollutant toxicity) but hardly addressable; most often, sequential or non-exhaustive extractions are used [55,56]. The potential of bioreporters remains here dominantly unexploited. The soil is a complex matrix with many potential interferences which need to be understood before bioreporter bioassays will become standard tools in environmental analytical chemistry. The route to validate bioavailability bioassay remains long.

## 5. Conclusions

We evaluated the performance of *mer-lux* bioluminescent bioreporter *Escherichia coli* ARL1 on real soil samples with variable mercury content and speciation. Using soil suspensions, water extracts, and laccase mediated extracts, the bioreporter has shown the capability to estimate semi-quantitatively the bioavailable mercury content in soil. Laccase extraction performed best from tested options. Significant matrix effects of humic acids and metals on bioluminescence were observed. Overall, the results show that the ARL1 bioassay can be used as a supplement to instrumental analyses for estimation of bioavailable mercury in soils. Nevertheless, more research on interferences and differently bound mercury species is needed.

## Figures and Tables

**Figure 1 sensors-20-03138-f001:**
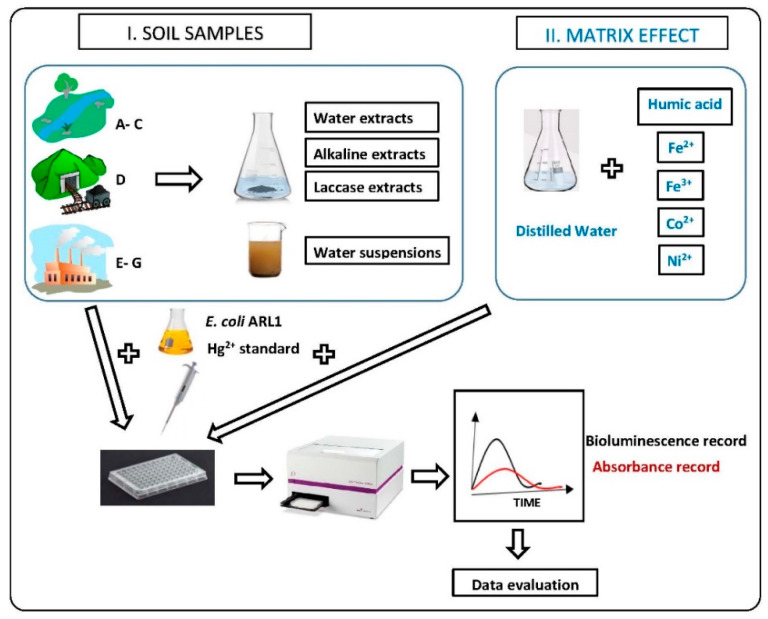
Schema of the *mer-lux* bioassay.

**Figure 2 sensors-20-03138-f002:**
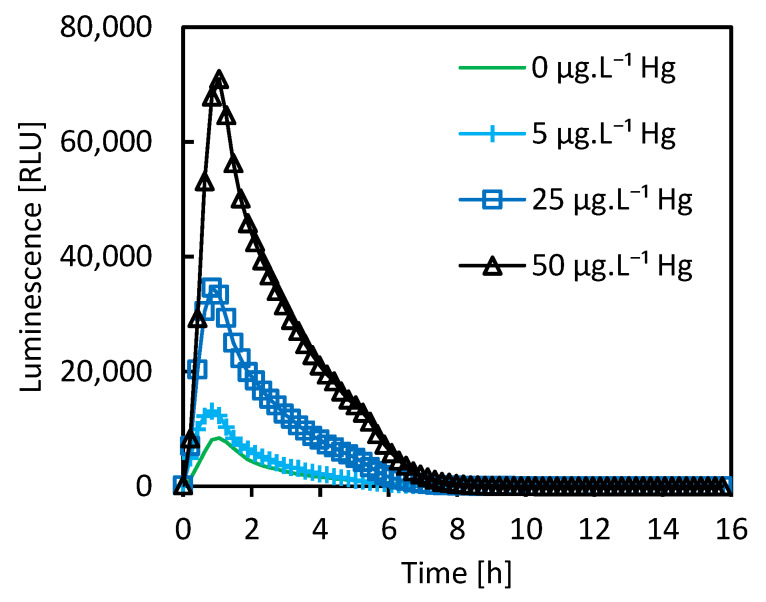
Time courses of measured *E. coli* ARL1 luminescence in soil suspensions of soil sample A with different standard additions of mercury (II) chloride.

**Figure 3 sensors-20-03138-f003:**
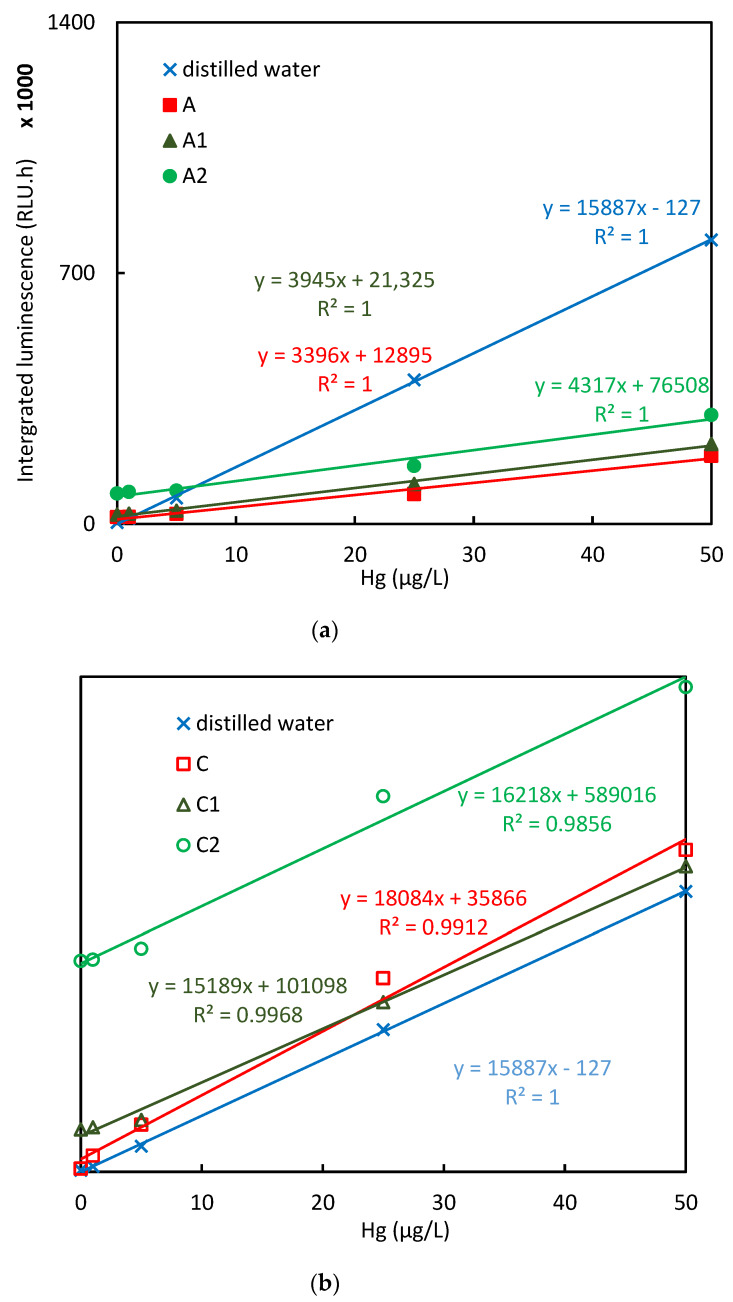
Integrated values of the measured luminescence induced in *E. coli* ARL1 by soil samples and distilled water as a function of standard additions of mercury (II) chloride expressed as 0; 1; 5; 25; 50 μg/L of mercury in final solution. (**a**) samples A, A1, A2; (**b**) samples C, C1, C2.

**Figure 4 sensors-20-03138-f004:**
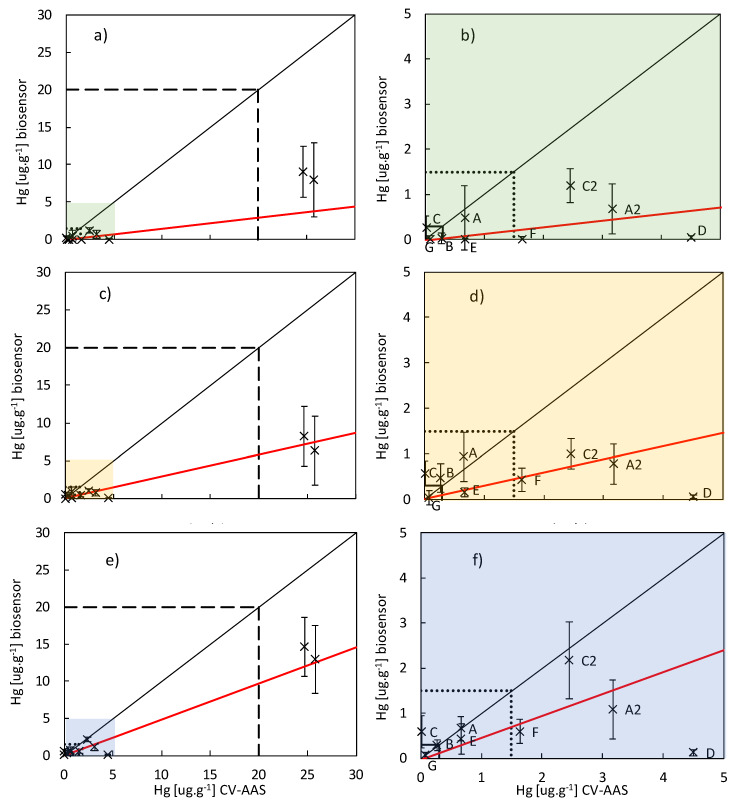
Each plot depicts Hg concentrations in soil samples (A–G, all were used in each plot) estimated by the bioreporter *E. coli* ARL1 (presumably bioavailable, Y axis) related to the total mercury concentrations assessed by CV AAS (X axis) for three different ways of sample preparations: plots (**a**,**b**)—soil suspensions, plots (**c**,**d**)—water extracts, plots (**e**,**f**)—laccase extracts. The diagonals represent theoretical position of the measured points, when bioavailability would be 100%. Solid line—“preventive value” 0.3 mg/kg, dotted line—“indication value” 1.5 mg/kg, dashed line “indication value” of 20 mg/kg. Presented points are averages from three to five independent repeats of the given pretreatment method and standard addition. Error bars represent sample standard deviations. For details of the regression curves see Table 4.

**Figure 5 sensors-20-03138-f005:**
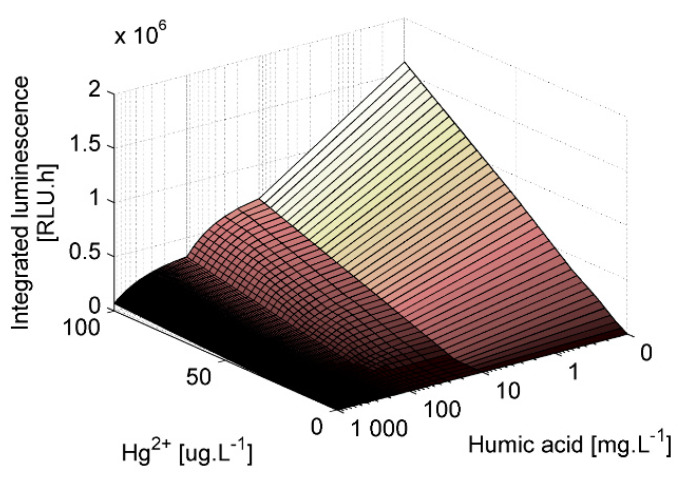
Dependence of integrated luminescence measured in water solutions on humic acids and mercury concentrations (mercury was added in the form of mercury (II) chloride).

**Figure 6 sensors-20-03138-f006:**
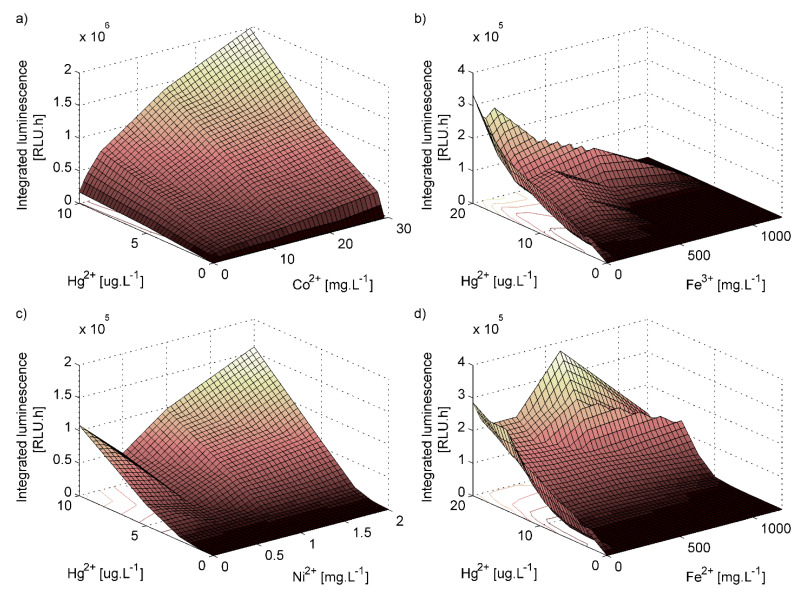
Dependence of integrated bioluminescence measured in water solutions on mercury concentrations (mercury was added in the form of mercury (II) chloride) under influence of added concentrations of chlorides of (**a**) cobalt (II), (**b**) iron (III), (**c**) nickel (II) and (**d**) iron (II).

**Table 1 sensors-20-03138-t001:** Detailed values and their statistics for mercury content determined using laccase extraction and bioreporter. *t*-test compared the significance of the value compared to 0.

Sample	TOC ± SE (%)	Hg Instr. ^1^ mg/kg ± SE	Bioactive Hg mg/kg ± SE ^2^	n	RSD Sample	*p*-Value of *t*-Test ^3^	Bioactive/Total Hg
A	33.6 ± 1.2	0.66 ± 0.03	0.66 ± 0.13	5	45%	0.008	99%
A1 *	33.6 ± 1.2	3.2 ± 0.2	1.1 ± 0.4	3	56%	*0.091*	34%
A2 *	33.6 ± 1.2	25.8 ± 1.3	12.8 ± 4.4	3	59%	*0.099*	50%
B	7.9 ± 0.3	0.27 ± 0.01	0.31 ± 0.09	3	51%	*0.076*	113%
C	0.4 ± 0.0	0.01 ± 0.00	0.56 ± 0.20	3	63%	*0.11*	3985%
C1 *	0.4 ± 0.0	2.5 ± 0.1	2.2 ± 0.5	3	43%	*0.055*	88%
C2 *	0.4 ± 0.0	24.7 ± 1.3	15.0 ± 3.1	3	36%	0.040	61%
D **	1.5 ± 0.1	4.5 ± 0.2	0.12 ± 0.04	3	57%	*0.094*	3%
E	11.3 ± 0.4	0.68 ± 0.03	0.43 ± 0.19	4	91%	*0.115*	63%
F	8.5 ± 0.3	1.64 ± 0.08	0.59 ± 0.15	4	52%	0.032	36%
G	8.1 ± 0.3	0.08 ± 0.00	0.08 ± 0.01	4	38%	0.013	93%

TOC—Total organic carbon; SE— standard error of the mean; RSD—Relative standard deviation of the repeated runs; * These soil samples were prepared by addition of HgCl_2_ (water solution) to the soil A or C respectively. ** Majority of the mercury presented in this soil was in the form of non-bioavailable cinnabar (HgS) [13]. ^1^ Hg instr.—concentration of Hg determined by AAS; ^2^ SE of the mean calculated from the n means of the repeated runs; ^3^ values insignificant (α = 0.05) are typed in italics.

**Table 2 sensors-20-03138-t002:** Spearman correlation between soil Hg concentrations determined by instrument analyses (CV-AAS), bioluminescence response of the bioreporter and total organic carbon. Significant correlations (α = 0.05 are typed bold-face).

	1.	2.	3.	4.	5.
1. Total organic carbon (%)	1.00				
2. Instrument analyses (CV AAS)	0.14	1.00			
3. Soil suspensions	−0.09	**0.65**	1.00		
4. Laccase extraction	0.06	**0.63**	**0.87**	1.00	
5. Water extraction	−0.01	0.53	**0.89**	**0.97**	1.00

**Table 3 sensors-20-03138-t003:** Specific growth rates of *E. coli* ARL1 in the presence of humic acid and mercury (II) chloride.

	Specific Growth Rates of *E. coli* ARL1 (h^−1^)				
Humic Acid	Hg 100 µg/L	Hg 50 µg/L	Hg 25 µg/L	Hg 12.5 µg/L	Hg 0 µg/L^−1^
1000 mg/L	0.20	0.20	0.20	0.20	0.21
100 mg/L	0.20	0.20	0.20	0.20	0.21
10 mg/L	0.19	0.19	0.19	0.20	0.21
0 mg/L	0.14	0.15	0.15	0.15	0.15

**Table 4 sensors-20-03138-t004:** Statistical evaluation of linear dependences (weighed regression) of Hg concentrations estimated by bioreporter on values determined by means of instrumental analysis; bold font indicates statistically significant results based on the hypothesis in the left column.

	Soil Suspension	Laccase Extraction	Water Extraction
(a) All samples			
Slope ± SDest	**0.033 ± 0.015**	0.019 ± 0.018	−0.028 ± 0.023
slope = 0?	**no (P = 0.040)**	yes (P = 0.28)	yes (P = 0.24)
slope = 1?	no (P < 10^−36^)	no (P < 10^−35^)	no (P < 10^−34^)
Intercept (mg/kg) ± SDest	−0.074 ± 0.032	0.093 ± 0.027	0.221 ± 0.090
intercept = 0?	**no (P = 0.025)**	**no (P = 0.0015)**	**no (P = 0.018)**
(b) Without proved biologically unavailable samples			
Slope ± SDest	**0.146 ± 0.051**	**0.487 ± 0.063**	**0.290 ± 0.046**
slope = 0?	**no (P = 0.0085)**	**no (P < 10^−7^)**	**no (P < 10^−6^)**
slope = 1?	no (P < 10^−14^)	no (P < 10^−8^)	no (P < 10^−15^)
Intercept (mg/kg) ± SDest	−0.160 ± 0.046	0.042 ± 0.018	0.019 ± 0.067
intercept = 0?	**no (P = 0.0018)**	**no (P = 0.027)**	yes (P = 0.774)

SDest—standard deviation of the estimate.

## Data Availability

Primary time-bioluminescence curves are available on the open-access server of Jan Evangelista Purkyně University (doi is under process of registration).

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
