# Peer review of "Estimation of Hg(II) in Soil Samples by Bioluminescent Bacterial Bioreporter E. coli ARL1, and the Effect of Humic Acids and Metal Ions on the Biosensor Performance"

_sensors, 2020, doi:10.3390/s20113138_

Round 1

Reviewer 1 Report

This MS investigates the suitability of using a prokaryotic biosensor (E. coli ARL1) to detect the bioavailable Hg content of soil samples. This follows on from the group’s previous work on Hg in atmospheric and water samples.

I really appreciated the MS’s frank discussion of the challenges of quantifying compounds from complex samples like soils, where organic and inorganic components almost always interfere with the ability to measure target molecules. I also really like the humility the authors have in acknowledging these challenges, and that given their results their proposed methods are at best semi-quantitative and must be paired with other Hg measurement methods, like the modified CV-AAS they cite and use here. Although this is mainly a paper with negative results, they are important to illustrate the challenges of using biosensors to monitor metals in soil and other complex mixtures such as wastewater, and should be published.

I have a few comments and suggestions that should be incorporated prior to publishing.

Lines 55-56 and 421-422: I’m unclear how they suggest Hg enters plants... they seem to suggest that significant amounts of Hg(0) vapor enters leaves during photosynthesis and is deposited in the leaf mesophyll cells. This seems unlikely, although it’s possible that this Hg reacts chemically within the substomal space and is deposited on the surface of the mesophyll cells. The main route by which Hg enters plant leaves is likely via xylem transport from the soil, like other dissolved minerals. This could result in deposition in leaves and roots; see, e.g., review by Azevedo and Rodriguez (2012) for details: https://www.hindawi.com/journals/jb/2012/848614/

. The authors should rewrite these lines for clarity given the uncertainty on how Hg is metabolized in plants. They may be confusing the terms transpiration and gas exchange.

The introduction could also benefit from a clearer brief discussion on the various charged forms Hg is found in soil (Hg0, Hg2+, etc.), and their relative abundance and toxicity. Perhaps comment on soil pH effects as well, as well as clarify whether the analytical technique chosen (CV-AAS) can distinguish between these... I think that it unfortunately cannot, based on the data and comments in the MS. If any of this information is known, it would clarify the arguments the authors make.

Other minor comments:

L 83: ...(HgS) is..., not in

L107: _The_ present paper...

L123-4: ...presence of divalent mercury cations (Hg2+).

Table 1: belongs in Results. Not referred to in text. Include +/- SD for Hg in soil column.

L179: ...it was neutralized...: to pH 7?

L221: _A_ series of experiments...

L289-90: ...Q-test; the data contained no outliers.

Fig 3: Clarify in the description whether all soil samples A to G were combined in these plots (e.g., each X is one of these samples).

Table 3: clarify whether “Hg instr.” Is measured using CV-AAS, and Hg measured using the bioreporter. Looks like the SE for the Hg instr. method is set at 5%; can you report the actual measured SD instead?

Table 4: Specific growth rates are presumably _OD600_  per h.

L388: _The_ presence...

L415: ...believed to respond to the bioavailable...

L435-6: Re ...responses to bioavailable mercury proportion only: Give some references indicating that this might be true, e.g., that Hg2+ is the only bioavailable Hg species.

Table 5: Remove () after “Slope”. What does the bolded text indicate?

L457: not “huge”... maybe “considerable”

L462: remove “literally”

L465... in most of the soils. Give a reference for this statement, or indicate whether you performed these measurements in the MS.

L481: remove “2010”

L483-4: ...20 to 42 times higher via the bioreporter than the total Hg content by direct measurement using CV-AAS.

L494-5: ... reliably determine whether a given sample exceeded or fulfilled...

Reviewer 2 Report

Dear Authors,
The article is interesting but it requires corrections and addition of supplemental information.

Major Concerns
1. How pH of the extracts and their mixtures with E. coli ARL1 culture was controlled? Acidity of the mixture may play a crucial role in bioluminescence, hence pH affects the measurement’s result.
2. In discussion of a metal ion influence on a measurement result, problem of possible ion hydrolysis was omitted. Particularly, concentrations of Fe(III) and Fe(II) were high enough for pH influence via hydrolysis and possible precipitation of Fe hydroxides solid phase. I suppose a deliberation of this mechanism in the mixtures studied. Consult, for example, Paul L. Brown and Christian Ekberg, Hydrolysis of Metal Ions, Verlag GmbH & Co.KgaA (2016), ePDF ISBN: 978-3-527-65621-9.
3. For assessment of null hypothesis concerning equality of means in 2 normally distributed population the t-test can be used. But the concentrations are not normally distributed. The RSD in Tab. 3 indicate that approx. 11% (for RSD=50%) of data in population are lower than 0, values invalid for concentrations.
Distribution of minor component concentration often follows lognormal function. I suppose testing logarythmised concentrations, not crude values.

Minor Concerns
1. Log-log scale will improve visibility of relationships in Fig. 3.
2. Number of significant figures (digits) should be appropriate to estimated precision of the data. For example, in Tab. 3, for sample A2 concentration ± standard error (SE) was 25.795±1.290. This form supposes very high accuracy of Hg determination, described by 5 meaning digits. But the standard error denies such conclusion. It is sufficient to round SE to 2 meaning digits and present a result with the same precision – 25.0±1.3.
3. The unit mg.kg -1 looks much better in the form mg/kg.

Author Response

Please see the attchment.

Reviewer 3 Report

The current manuscript reports advancement in biological sensing of mercury in soil samples using whole-cell bacterial bioreporter Escherichia coli ARL1. The presence of Hg(II) in several soil samples was determined using different soil pretreatments for quantification of bioassay performance. The results show interesting trends of the bioassay performance with different matrix constituents. The following suggestions are provided in the spirit of improving the manuscript:

  1. The time course profiles shown in Figure 1 requires additional explanation. For example, why is the profile of the curves in the observed shape? What are the implications of various regimes of the curve in the time axis?
  2. Why are two plots presented for each sample preparation method in Figure 3? The authors should condense them into one plot for each method and use the space to show the magnified version of the low concentration region where the linearity is not apparent in its current form. Further, the authors stated, “For the majority of soil samples, the biosensor estimated values were significantly lower comparing to the total ones and they represent 1% to 73% of the total value.” The authors should consider probing into potential relation between the biosensors values and CV-AAS values to obtain something similar to the calibration curve. In the current plots, it is difficult to see anything in the low concentration region. It must be rectified.
  3. The authors should elucidate the results in Figures 4 and 5 in more detail in the results section at appropriate places.
  4. While the discussion section provides a detailed discussion of the observed results, it lacks appropriate references to the Figures and Tables where these observations were made. This makes the discussion section difficult to follow. Also, it will allow a reader interested in probing the results shown in the Figures further, to easily find the related discussion. The authors should, therefore, make these changes to improve the readability.
  5. The authors should provide a Figure to summarize the experiments conducted in the manuscript and provide a study schematic for preserving the big picture of the manuscript.
  6. The discussion section should include a discussion on the utility of the current results and pathways for motivating future work in the area.
  7. Finally, the language in the manuscript needs to be improved to correct grammatical and typographical errors.

Author Response

Please see the attchment.

Reviewer 4 Report

Overall, this is quite a novel study and an interesting attempt at assessing the bioavailability of Hg in soils using a genetically modified bioluminescent bacteria: Escherichia coli ARL1. The background of the study and methods are generally well described and I believe it has merits for publication. Nonetheless, there are some short-comings that need to be address and are summarised below. Once these points are addressed I feel the manuscript merits publication, especially due to the extensive efforts and quality control placed on the experimental design.

GENERAL COMMENTS:

  1. In the current form of these bacteria and based on these results, they are not very supportive of Escherichia coli ARL1 being an appropriate bioavailable Hg indicator in soils when the full range of these results are considered even focussing on the laccase extracts. This finding is fine in itself and is publishable. However, there does appear to be some overreach on the effectiveness of these bacteria for this purpose. Thus, I suggest some of the conclusions and suggestions of the application of these bacteria be toned down. Much more work is required before they could be viably applied to perform any sort of health assessment when actual lives and livelihoods are at stake, especially considering the variability of natural soils and that they can be high in non-bioavailable forms of Hg and/or other metal ions and/or organic substances that may affect results. Abstract and conclusions must reflect these changes.
  2. Some of the data presentations need to be improved and this is generally reflective of point 1. Proper reporting of figures and tables will more appropriately describe and display the trends of the data.

SPECIFIC COMMENTS:

Line 48: Change "moving" to "transfer between"

Line 51: "inputs" should be "primary inputs" (important to distinguish this). Just like carbon secondary inputs are much higher than the primary inputs, but the primary inputs are what perturbs the cycle.

Lines 55-57: References needed.

Line 73: I would suggest using MeHg instead of CH3Hg+ as the acronym for methylmercury. Technically this is mono-methyl mercury, and you are not accounting for di-methyl mercury by using this formula. If you prefer to use CH3Hg+ then you should write mono-methyl mercury here.

Line 73: Remove "the" before methylmercury

Lines 83-84: please change "...non-available; opposite to this is HgCl2, which is..."

Line 85: Please change "speciation..." to "Speciation or fractionation..."

Line 86: a sentence on exactly why speciation [or fractionation] is not sufficient is important here. Something like "because there are large uncertainties surrounding the make-up of the species or fractions that can be identified"...something along these lines...

Line 87: "from its nature" should be "by their nature"

Line 87: Please change "This complete task can" to "This complete task may"...it's less assertive and more appropriate given that is the research question and hypothesis [essentially].

Line 89: "comparing" to "compared".

Line 90: Does it have to be so specific to the Czech Republic? if it is in compliance with the more general EU legislation then I feel that is a more appropriate benchmark to follow.

Line 93-95: both these values listed in these two sentences are written as the "indication value". So which is the actual indication value?

Line 111: Change "aiming" to "aimed at" or "focussed on".

Line 131: End parenthesis ")"

Line 138: The word "soils" is missing from "In brief, [soils] were..."

Line 142: What about standard reference materials? What were run during the CV-AAS analysis and what were the recoveries? That is VERY important and must be listed here.

Lines 143 & 151: "...consistently <5%" - the authors should be a little more quantitative here. Report the mean and variability of these standard deviations - essentially the RSD%

Lines 176: "stock solution was..." - what is the reported uncertainty of this value.

Line 287: standard deviations of "139±142%, 54±15% and 74±24%" are quite high and show very large variability and inconsistency of results and method. This in itself isn't very good for the effectiveness of the bacteria in determining the mercury bioavailability.

Lines 310-311: I don't see why these alkaline results wouldn't be included in the table 3 at least, just to show that it was not functional.

Line 314: Change "on the Figure 1" to "in Figure 1"

Lines 315-316: Change "...case of the soil labelled A suspensions..." to "case of the suspensions for soil sample A..."

Line 316: Change "way of measurement" to "method of measurement"

Lines 316-317: These data should be presented in a supplementary information section. Let the readers decide whether or not they are similar - and if they are then you should have no problem showing these data in such a supplement. Open Source!

Lines 324: what are the "total ones"? Total Hg concentrations? Please write that.

Figure 2: I think this would be a better figure if you scaled the axes the same to show the difference more clearly between soils.

Figure 3: This figure needs a lot of improvement. (i) label the different soil types/samples instead of grouping them together - perhaps with different letters instead of X for all samples? (ii) Why is there no regression analysis of these data? Fit the data with a regressions, equation, R2 value, and p-value (significance)? (iii) What is the difference between plot e and plot f; plot a and plot b; and plot c and plot d - what is the difference between the left and right columns of figure panels?

Line 355: Please provide this data in a supplement.

Lines 389: "substantially", was this tested statistically or is this a subjective remark?

Lines 407-410: This is HUGELY confusing. Were the slopes significant using regression analysis? Table 5 could more or less just be included in Figure 4. Figure 4 should include the slopes of these graphs, their R2 values and p-values. They really do not appear correlated as the previous sentences suggest.

Lines 415-417: Given these are controlled experiments, with different treatments then increased Hg should also mean increased bioavailable Hg too (in most cases too), but the results state these bacteria cannot pick up those changes.

Line 419: Sequential extractions are notoriously problematic for Hg and misdescribe fractions (Nirel and Morel, 1990; Biester and Scholz, 1996; Brocza et al., 2019).

Lines 419-422: Why? wouldn't the Hg that is most soluble be the most relevant to human health as it can be incorporated into runoff, end up in sediments and become methylated? If it's evaded and then taken up again by plants and then deposited again as literfall it will mostly just return to the soil. Unless it's MeHg it won't bioaccumulate. Are the authors suggesting uptake into leaves of agricultural plants then consumed by humans is a health concern - which I am certain is not (never have I heard consumption of plants that take up Hg0 via stomata to be a health concern).

Line 427: I don't agree with this assessment. Simple pyrolytic thermal desorption can even be done with DMA80-type instruments (those used for the concentration analysis in this study)and that alone would have greatly assisted the results here (Windmöller et al., 2017).

Lines 430-432: These should be plotted as a figure. Although this is an interesting finding and should be discussed it should also mention that being this selective with "viable" soils is a severe limitation of the method's application.

Line 447: Change "has" to "have"

Lines 461-465: Further evidence against the application of the method to actual soils that have highly variable concentrations of metal ions. This again should be a big reason for toning down the recommendations based on these data.

Lines 469-471: This should be referenced. Humic acids and other organic phases are known to sorb Hg and reduce its bioavailability

Lines 505-508: I do not believe this is semi-quantitative at all, at best qualitative, but in it's current methodological application, really it does not seem highly suitable to soils and this should not be overplayed. 

Windmöller, C. C., Silva, N. C., Andrade, P. H. M., Mendes, L. A., & do Valle, C. M. (2017). Use of a direct mercury analyzer® for mercury speciation in different matrices without sample preparation. Analytical Methods9(14), 2159-2167.

Author Response

Please see the attchment.

Round 2

Reviewer 2 Report

No comments